# Synthesis of Carvone-Derived 1,2,3-Triazoles Study of Their Antioxidant Properties and Interaction with Bovine Serum Albumin

**DOI:** 10.3390/molecules23112991

**Published:** 2018-11-16

**Authors:** Armen S. Galstyan, Armen I. Martiryan, Karine R. Grigoryan, Armine G. Ghazaryan, Melanya A. Samvelyan, Tariel V. Ghochikyan, Valentine G. Nenajdenko

**Affiliations:** 1Faculty of Chemistry, Yerevan State University, 1 Alex Manoogian Str., Yerevan 0025, Armenia; armart@ysu.am (A.I.M.); kara@ysu.am (K.R.G.); armghazaryan@ysu.am (A.G.G.); msamvelyan@ysu.am (M.A.S.); ghochikyan@ysu.am (T.V.G.); 2Department of Chemistry, M.V. Lomonosov Moscow State University, 119991 Moscow, Russia

**Keywords:** carvone, 1,2,3-triazole, click chemistry, acetylene, antioxidant activity, bovine serum albumin

## Abstract

Natural L-carvone was utilized as a starting material for an efficient synthesis of some terpenyl-derived 1,2,3-triazoles. Chlorination of carvone, followed by nucleophilic substitution with sodium azide resulted in the preparation of 10-azidocarvone. Subsequent CuAAC click reaction with propargylated derivatives provided an efficient synthetic route to a set of terpenyl-derived conjugates with increased solubility in water. All investigated compounds exhibit high antioxidant activity, which is comparable with that of vitamin C. It was also found that serum albumin and the terpenyl-1,2,3-triazoles hybrids spontaneously undergo reversible binding driven by hydrophobic interactions, suggesting that serum albumin can transport the target triazoles.

## 1. Introduction

Natural compounds play a significant role in the design of new drugs and prevention of various diseases [1]. Thus, more than 60% of current drugs for the treatment of cancer and infectious diseases are of natural origin [2]. Terpenoids are one of the largest classes of chiral natural compounds, which includes more than 23,000 compounds. Many terpene derivatives are widely used in perfumes, playing the role of cosmetic products and food additives [3]. However, medicine is not less important area of application of terpenes, since the majority of compounds of this class exhibits pronounced biological activity. For example, the antimalarial drug artemisinin and the anticancer drug paclitaxel (Taxol^®^) are the two most prominent members of the class of terpenes used in medicine (Figure 1). 

Annual sales of terpenic derivatives in the world are estimated at about $20 billion [4]. The scope of terpenes is constantly expanding; therefore, terpenoids will play an increasingly important role as therapeutic and prophylactic agents for human diseases.

Monoterpenes–C_10_H_16_ compounds and their derivatives are the simplest type of terpenoids. They are the basis of essential oils, floral aromas and perform plant protective functions [5,6]. Despite the fact that these are small and fairly simple molecules, a number of monoterpenes have an antitumor effect, demonstrating not only the ability to prevent the formation or progression of cancer, but the ability to reduce existing malignant tumors [7]. Most abundant in nature are terpenoids having menthane skeleton (Figure 2). For example, limonene is the most common monocyclic monoterpene found in nature. It can be found in a variety of trees and grasses (for example, *Mentha* spp.). Limonene is an important component of the peel of oranges and lemons, as well as the essential oil of cumin. The limonene fragment can be found in the structure of many drugs [7,8,9,10,11]. Carvone, the main monoterpene of cumin seed. It has been shown that carvone oil prevents chemically induced carcinoma of the lung [12]. In addition, carveol has chemoprophylactic activity against breast cancer during the initiation phase [13]. Perillyl alcohol, a hydroxylated limonene analog, exhibits chemopreventive activity against chemically induced liver cancer [14] and tumor recurrences in animal models [15].

Nowadays, one of the most promising synthetic trends for expanding molecular complexity from a given scaffold have been the “click” reactions. “Click” reactions are a class of efficient, fast, universal, and selective reactions that are characterized by high yields and simple isolation of products. The standard “click” of chemistry for today is the formation of 1,2,3-triazoles catalyzed by copper salts (I) in the reaction of azides and terminal alkynes (CuAAC) [16,17,18,19,20]. Although 1,2,3-triazole fragment is generally absent in natural compounds, synthetic molecules containing 1,2,3-triazole cycles represent a wide class of physiologically active substances exhibiting various types of biological activity, for example, antibacterial, antiallergic, antiviral, antifungal properties [21,22]. Apparently, the pharmacological activity of compounds containing 1,2,3-triazole groups is due to the structural and electronic similarity of the 1,2,3-triazole fragment and the amide group. The 1,2,3-triazole fragment can be considered as a conformationally constrained bioisostere of the amide group. Greater stability of the 1,2,3-triazole scaffold under physiological conditions, led to the widespread use of the “click” reaction as an effective method for the synthesis of various biologically active compounds in drug design. The triazole fragment is an important pharmacophoric unit, and a large number of drugs containing this heterocycle are known (Figure 3). Due to the commercial success of some pharmaceutical preparations based on the triazole ring, many pharmaceutical companies and academic groups have shown interest in developing new methods for synthesizing triazole compounds and screening their biological activity [23,24]. For example, antifungal drugs containing triazole rings are known: itraconazole, fluconazole, voriconazole [25], antiviral drug ribavirin and murbritinib (used to treat breast, bladder, kidney and prostate cancer). Ribavirin is a drug for the treatment of viral infections such as herpes and hepatitis [26,27].

## 2. Results and Discussion

This study is devoted to the synthesis of carvone-derived 1,2,3-triazoles **4a**–**i** prepared according to the strategy outlined in Scheme 1. We proposed that due to the presence of the enone fragment in the structure, such conjugates would behave as antioxidants. On the other hand, the triazole and amine moieties will play an auxiliary role of improving water solubility of these molecules. 10-Azido-carvone was chosen as a key building block to study copper-catalyzed azide alkyne cycloaddition (CuAAC). A number of propargylated amino derivatives **3** were studied as alkyne partners for this reaction. They can be prepared using standard alkylation of NH-derivatives with propargyl bromide (Scheme 1) [28].

To achieve this goal, L-carvone (**1**) was chlorinated using calcium hypochlorite−CO_2_ system to provide 10-chlorocarvone (**2**) [29]. Subsequent treatment of **2** with sodium azide resulted in synthesis of 10-azidocarvone. We decided to utilize synthetically attractive one-pot protocol to avoid isolation and purification of this intermediate product. For this aim, chloride **2** was treated with sodium azide in DMSO or acetonitrile to yield 10-azido-carvone, which was used without isolation in model CuAAC reaction with *N*-propargylmorpholine **3a**. It was found that the in the case of DMSO as a solvent the yield of model conjugate **4a** is moderate. However, with acetonitrile as a solvent and copper iodide as a catalyst, the isolated yield was improved significantly. For example, **4a** was obtained in up to 78% yield within a reasonable reaction time (10 h) using only 5 mol % of copper iodide (Table 1). Probably, such observation can be explained by formation of the complex of copper iodide with final products.

Next, the reaction with a number of propargylated amines and amides was investigated. To our delight, the corresponding triazole derived carvone conjugates **4** were isolated in up to 84% yield. The efficient procedure provided **4** as crystalline compounds. Their structure was confirmed by combination of spectroscopic methods (see SI). As a result, this part of study gave as a family of carvone derivatives **4** with variable amino- (amido) substituents in the triazole ring (Table 2).

With these new compounds in hand, we decided to study the antioxidant properties of the synthesized terpenyl-1,2,3-triazoles **4**. For this aim the interaction with HO**^•^** radicals in the presence of the competitive acceptor 4-nitroso-*N*,*N*-dimethylaniline (PNDMA) was studied [30,31,32]. The initiation of HO^•^ radicals was carried out by photolysis of H_2_O_2_ (10^−3^ mol·L^−1^) under the action of UV radiation (λ = 313 nm with the use of a special filter). The rate of initiation of HO^•^ radicals (Figure 4a–f) was determined by the change in PNDMA absorption (А_440_). It was found that compounds **4a**–**e** have better water-solubility and their antioxidant properties were investigated. Below are given the kinetic data of the effect of **4а**–**e** different concentrations on the optical density of PNDMA, depending on the time of irradiation (Figure 4a–e).

The constants of the interaction rate of HO^•^ radicals with **4a**–**e** were calculated by the Equation (1) [31,32]: k_OH + P_ = 1.25 × 10^10^([PNDMA]/[P]) × [(W_1_/W_2_) − 1], mol^−1^ s^−1^ L(1)
where 1.25 × 10^10^ mol^−1^ s^−1^ L is the constant of the interaction rate of HO^•^ radicals with PNDMA, [P] is the concentration of **4a**–**e**, W_1_ and W_2_ are rates of PNDMA discoloring in distilled water and in the presence of **4a**–**e** respectively. The rate constants are shown in Table 3. As one can see from Table 3, all the compounds tested exhibit significant antioxidant activity. Moreover compound **4b** is slightly inferior to the known antioxidant ascorbic acid used as a control compound.

The transport of active aglycones of drugs is known to be carried out by various interactions with the proteins. The most abundant protein in the blood plasma is the transport protein serum albumin (up to 60%) which can reversibly bind various endogenous and exogenous compounds [33,34,35]. To reveal possibility of such binding, fluorescence spectroscopy was used for model compound **4e** with the bovine serum albumin (BSA). The thermodynamic parameters (ΔH, ΔS and ΔG) can be used to propose the binding mode. For the typical hydrophobic interactions, both ΔH and ΔS are positive, while negative ΔH and ΔS result from the hydrogen bond formation and van der Waals forces, and electrostatic interactions are responsible for the cases when ΔH < 0 and ΔS > 0 [36]. The value of binding constant (K_b_) for transport protein ligand interaction in the range of10^3^–10^6^ M^−1^ indicate the reversibility of binding [37]. The results obtained for BSA-**4e** system are summarized in Table 4. Negative values of ΔG show that the binding process proceeds spontaneously. The positive values of ΔH and ΔS show that the stability of the BSA-**4e** system is due mainly to hydrophobic interactions.

The distance between the protein and the ligand can be calculated using the theory of resonance energy transfer (Foerster theory) [38]. Average distance between BSA-**4e** decreases in the range of 2–8 nm and the energy transfer efficiency increases when the temperature was increased from 298 K to 308 K (Table 5). However, such interactions of BSA with **4e** are not very strong. Thus, the transport of **4e** can be performed by serum albumin-BSA.

## 3. Materials and Methods

### 3.1. General Information

^1^H- and ^13^C-NMR spectra were recorded on a Mercury-300 MHz instrument (Varian, Palo Alto, CA, USA) in DMSO-CCl_4_ mixture (1:3) or on an AVANCE 400 MHz spectrometer (Bruker, Billerica, MA, USA) in CDCl_3_. Chemical shifts (δ) in ppm are reported as quoted relative to the residual signals of chloroform-*d* (7.26 for ^1^H-NMR and 77.16 for ^13^C-NMR) or DMSO-*d*_6_ (2.50 for ^1^H-NMR and 39.52 for ^13^C-NMR) as internal references. The coupling constants (*J*) are given in Hertz. ESI-MS spectra were measured with a MicroTof instrument (Bruker Daltonics, Billerica, MA, USA). TLC analysis was performed on Xtra SIL G/UV_254_ plates (Macherey-Nagel, Düren, German). All reagents were of reagent grade and were used as such or distilled prior to use. Compounds **2** and **3a**–**i** were synthesized according to the procedure described in [28,29]. Melting points were determined on a Boetius micro-heating stage and a SMP10.

*(5R)-5-(3-chloroprop-1-en-2-yl)-2-methylcyclohex-2-en-1-one.* Yield 67%, b.p. 110–116 °C/1 Torr, $n_{D}^{20}$ 1.5285, ${\left[\alpha \right]}_{D}^{20}$ −44.05 (C 2.17 DCM).

*4-(Prop-2-yn-1-yl)morpholine* (**3a**). Yield 53%, b.p. 71–72 °C/15 Torr, $n_{D}^{20}$ 1.4741. ^1^H-NMR (300 MHz, DMSO/CCl_4_—1/3) δ 3.63–3.56 (m, 4H, OCH_2_), 3.22 (d, *J* = 2.5 Hz, 2H, NCH_2_C≡CH), 2.51 (t, *J* = 2.5 Hz, 1H, ≡CH), 2.49–2.44 (m, 4H, NCH_2_). ^13^C-NMR (75 MHz, DMSO/CCl_4_—1/3) δ 78.0(C≡CH), 73.9(C≡CH), 65.8 (OCH_2_), 51.3 (NCH_2_), 46.4(NCH_2_C≡).

*1-Methyl-4-(prop-2-yn-1-yl)piperazine* (**3b**). Yield 49%, b.p. 74 °C/12 Torr, $n_{D}^{20}$ 1.4799. ^1^H-NMR (300 MHz, DMSO/CCl_4_—1/3) δ 3.20 (d, *J* = 2.5 Hz, 2H, NCH_2_C≡CH), 2.55–2.42 (m, 5H, NCH_2_, ≡CH), 2.42–2.21 (m, 4H, NCH_2_), 2.18 (s, 3H, NCH_3_). ^13^C-NMR (75 MHz, DMSO/CCl_4_—1/3) δ 78.4(C≡CH), 73.5(C≡CH), 54.2(NCH_2_), 50.8(NCH_2_), 46.1(NCH_2_C≡), 45.4(NCH_3_).

*1-(Prop-2-yn-1-yl)pyrrolidine* (**3c**). Yield 41%, b.p. 42–43 °C/17 Torr, $n_{D}^{20}$ 1.4590. ^1^H-NMR (300 MHz, DMSO/CCl_4_—1/3) δ 3.33 (d, *J* = 2.4 Hz, 2H, NCH_2_C≡CH), 2.58–2.50 (m, 4H, NCH_2_), 2.38 (t, *J* = 2.4 Hz, 1H, ≡CH), 1.82–1.70 (m, 4H, CH_2_). ^13^C-NMR (75 MHz, DMSO/CCl_4_—1/3) δ 78.9(C≡CH), 72.7(C≡CH), 51.2(NCH_2_), 41.8(NCH_2_C≡), 23.2(CH_2_).

*1-(Prop-2-yn-1-yl)piperidine* (**3d**). Yield 50%, b.p. 55–57 °C/12 Torr, $n_{D}^{20}$ 1.4706. ^1^H-NMR (300 MHz, DMSO/CCl_4_—1/3) δ 3.18 (d, *J* = 2.4 Hz, 2H, NCH_2_C≡CH), 2.48–2.35 (m, 5H-NCH_2_, ≡CH), 1.62–1.50 (m, 4H, CH_2_), 1.47–1.33 (m, 2H, CH_2_). ^13^C-NMR (75 MHz, DMSO/CCl_4_—1/3) δ 78.6(C≡CH), 73.2(C≡CH), 52.1(NCH_2_), 46.9(NCH_2_C≡), 25.2(CH_2_), 23.4(CH_2_).

*1-(Prop-2-yn-1-yl)azepane* (**3e**). Yield 55%, b.p. 71–72 °C/15 Torr, $n_{D}^{20}$ 1.4791. ^1^H-NMR (300 MHz, DMSO/CCl_4_—1/3) δ 3.28 (d, *J* = 2.4 Hz, 2H, NCH_2_C≡CH), 2.66–2.57 (m, 4H, NCH_2_), 2.35 (t, *J* = 2.4 Hz, 1H, ≡CH), 1.69–1.54 (m, 8H, (CH_2_)_4_). ^13^C-NMR (75 MHz, DMSO) δ 79.6(C≡CH), 72.3(C≡CH), 54.0(NCH_2_), 47.4(NCH_2_C≡), 27.8(CH_2_), 26.2(CH_2_).

*1-(Prop-2-yn-1-yl)pyrrolidine-2,5-dione* (**3f**). Yield 64%, b.p. 127 °C/2 Torr, $n_{D}^{20}$ 1.5124. ^1^H-NMR (300 MHz, DMSO/CCl_4_—1/3) δ 4.13 (d, *J* = 2.5 Hz, 2H, NCH_2_), 2.69 (s, 4H, CH_2_C=O), 2.60 (t, *J* = 2.5 Hz, 1H, ≡CH). ^13^C-NMR (75 MHz, DMSO/CCl_4_—1/3) δ 175.0 (C=O), 77.1(C≡CH), 71.8(C≡CH), 27.7 CH_2_C=O), 26.8 (NCH_2_).

*1-(Prop-2-yn-1-yl)-1H-indole-2,3-dione* (**3g**). Yield 64%, m.p. 162–163 °C (EtOH/H_2_O—1/1). ^1^H-NMR (300 MHz, DMSO/CCl_4_—1/3) δ 7.66 (td, *J* = 7.8, 1.4 Hz, 1H_arom_), 7.57 (ddd, *J* = 7.4, 1.3, 0.4 Hz, 1H_arom_), 7.22–7.12 (m, 2H_arom_), 4.53 (d, *J* = 2.5 Hz, 2H, NCH_2_), 2.81 (t, *J* = 2.5 Hz, 1H, ≡CH). ^13^C-NMR (75 MHz, DMSO/CCl_4_—1/3) δ 181.7(C=O), 156.3(NC=O), 149.3(C_arom_), 137.5(HC_arom_), 124.2(HC_arom_), 123.0(HC_arom_), 117.2(C_arom_), 110.8(HC_arom_), 76.1(C≡CH), 73.8(C≡CH), 28.7 (NCH_2_).

*2-(Prop-2-yn-1-yl)-1H-isoindole-1,3(2H)-dione* (**3h**). Yield 63%, m.p. 146–148 °C (EtOH). ^1^H-NMR (300 MHz, DMSO/CCl_4_—1/3) δ 7.89–7.84 (m, 2H_arom_), 7.83–7.78 (m, 2H_arom_), 4.37 (d, *J* = 2.5 Hz, 2H, NCH_2_), 2.64 (t, *J* = 2.5 Hz, 1H, ≡CH). ^13^C-NMR (75 MHz, DMSO/CCl_4_—1/3) δ 165.7(C=O), 133.7(HC_arom_), 131.4(C_arom_), 122.9(HC_arom_), 77.2(C≡CH), 72.2(C≡CH), 26.2(NCH_2_).

*1-(Prop-2-yn-1-yl)-1H-benzotriazole* (**3i**). Yield 45%, m.p. 62–63 °C (PhH/Hex—1/1). ^1^H-NMR (300 MHz, DMSO) δ 8.00 (dt, *J* = 8.3, 1.0 Hz, 1H_arom_), 7.82 (dt, *J* = 8.3, 1.0 Hz, 1H_arom_), 7.52 (ddd, *J* = 8.2, 6.9, 1.0 Hz, 1H_arom_), 7.37 (ddd, *J* = 8.2, 6.9, 1.0 Hz, 1H_arom_), 5.58 (d, *J* = 2.6 Hz, 2H, NCH_2_), 3.04 (t, *J* = 2.6 Hz, 1H, ≡CH). ^13^C-NMR (75 MHz, DMSO/CCl_4_—1/3) δ 145.3(C_arom_), 132.0(C_arom_), 126.7(HC_arom_), 123.2(HC_arom_), 119.1(HC_arom_), 109.9(HC_arom_), 75.9(C≡CH), 75.6(C≡CH), 37.1(NCH_2_).

*(5R)-5-(3-azidoprop-1-en-2-yl)-2-methylcyclohex-2-en-1-one* (**10-N_3_-Car.**). 369 mg (2 mmol) of (5*R*)-5-(3-chloroprop-1-en-2-yl)-2-methylcyclohex-2-en-1-one, 2 mL of acetonitrile, 156 mg (2.4 mmol) of sodium azide are placed in a round-bottomed 5 mL flask and heated with stirring for 10 h (60 °C). The mixture was poured into water (30 mL) and extracted with DCM (3 × 10 mL). The combined extracts were dried over Na_2_SO_4_, the volatiles were evaporated and the residue was purified via column chromatography on silica gel using DCM–Hex mixture (1:1) as an eluent. Yield 70%. ^1^H-NMR (300 MHz, DMSO/CCl_4_—1/.3) δ 6.74 (ddq, *J* = 5.5, 2.7, 1.3 Hz, 1H, HC=), 5.15 (s, *J* = 4.0 Hz, 1H^a^, H_2_C=), 5.09 (s, 1H^b^, H_2_C=), 3.87 (s, 2H, CH_2_N_3_), 2.93–2.66 (m, 1H^a^, CH_2_C=O), 2.61–2.44 (m, 2H, CH, CH_2_ in carbocycle), 2.43–2.22 (m, 2H, CH_2_ in carbocycle), 1.73 (dt, *J* = 2.6, 1.4 Hz, 3H, CH_3_). ^13^C-NMR (75 MHz, DMSO/CCl_4_—1/3) δ 196.5(C=O), 144.7, 142.8, 134.6, 113.4, 53.8, 42.2, 38.4, 30.6, 15.1(CH_3_).

### 3.2. General Method for the Preparation of Terpenyl-1,2,3-triazoles ***4a**–**i***

369 mg (2 mmol) of (5*R*)-5-(3-chloroprop-1-en-2-yl)-2-methylcyclohex-2-en-1-one, 2 mL of acetonitrile, 156 mg (2.4 mmol) of sodium azide are placed in a round-bottomed 5 mL flask and heated with stirring for 10 h (60 °C). After cooling, 242 mg (2.4 mmol) of triethylamine, 19 mg (0.1 mmol) of CuI and 2.4 mmol of the corresponding propargyl derivative are added to the resulting mixture. The whole is heated then for 10 h at 60 °C and the solvent is distilled off. The cooled mixture is dissolved in benzene, filtered and hexane is added to the filtrate. The precipitated crystals are filtered off, washed with hexane and dried.

*(5R)-2-methyl-5-(3-{4-[(morpholin-4-yl)methyl]-1H-1,2,3-triazol-1-yl}prop-1-en-2-yl)cyclohex-2-en-1-one* (**4a**). Yield 78%, m.p. 81–82 °C, ${\left[\alpha \right]}_{D}^{20}$ −31.65 (C 1.15 DCM). ^1^H-NMR (400 MHz, CDCl_3_) δ 7.47 (s, 1H, NCH), 6.69 (ddd, *J* = 5.8, 2.6, 1.4 Hz, 1H, HC=CCO), 5.13 (s, 1H^a^, H_2_C=), 5.06 (s, 1H^b^, H_2_C=), 5.04 (d, *J* = 15.8 Hz, 1H^a^, H_2_C=C-CH_2_N), 4.93 (d, *J* = 15.3 Hz, 1H^a^, H_2_C=C-CH_2_N), 3.75–3.68 (m, 4H, O(CH_2_CH_2_)_2_NCH_2_), 3.67 (s, 2H, O(CH_2_CH_2_)_2_NCH_2_), 2.65–2.43 (m, 7H, CH_2_C=O, O(CH_2_CH_2_)_2_N, CH in carbocycle), 2.42–2.18 (m, 2H, CH_2_ in carbocycle), 1.76 (dt, *J* = 2.5, 1.3 Hz, 3H, CH_3_). ^13^C-NMR (75 MHz, DMSO/CCl_4_—1/3) δ: 196.5 (C=O), 145.6, 143.0, 142.9, 134.5, 123.1, 113.3, 65.9, 52.9, 52.6, 52.5, 42.1, 37.9, 30.4, 15.1. HRMS (ESI) *m*/*z*: [M + H]^+^ Calcd for C_17_H_25_N_4_O_2_^+^ 317.1978, Found 317.1972.

*(5R)-2-methyl-5-(3-{4-[(4-methylpiperazin-1-yl)methyl]-1H-1,2,3-triazol-1-yl}prop-1-en-2-yl)cyclohex-2-en-1-one* (**4b**). Yield 79%, m.p. 117 °C, ${\left[\alpha \right]}_{D}^{20}$ −38.79 (C 1.044 DCM). ^1^H-NMR (400 MHz, CDCl_3_) δ 7.45 (s, 1H, NCH), 6.67 (ddd, *J* = 5.7, 2.7, 1.4 Hz, 1H, HC=CCO), 5.10 (s, 1H^a^, =CH_2_), 5.04 (s, 1H^b^, =CH_2_), 5.01 (d, *J* = 15.5 Hz, 1H^a^, H_2_C=C-CH_2_N), 4.90 (d, *J* = 15.2 Hz, 1H^b^, H_2_C=C-CH_2_N), 3.66 (s, 2H, CH_2_N(CH_2_CH_2_)_2_NCH_3_), 2.65–2.30 (m, 12H, CH and CH_2_ in carbocycle, CH_2_N), 2.25 (s, *J* = 7.6 Hz, 4H, CH_2_ in carbocycle, NCH_3_), 1.74 (dt, *J* = 2.5, 1.3 Hz, 3H, CH_3_). ^13^C-NMR (101 MHz, CDCl_3_) δ 198.5 (C=O), 145.2, 145.1, 143.8, 135.8, 122.6, 115.5, 55.1, 54.0, 53.4, 53.0, 46.1, 42.8, 38.3, 31.2, 15.7. HRMS (ESI) *m*/*z*: [M + H]^+^ Calcd for C_18_H_28_N_5_O^+^ 330.2294, Found 330.2293.

*(5R)-2-methyl-5-(3-{4-[(pyrrolidin-1-yl)methyl]-1H-1,2,3-triazol-1-yl}prop-1-en-2-yl)cyclohex-2-en-1-one* (**4c**). Yield 73%, m.p. 69–70 °C, ${\left[\alpha \right]}_{D}^{20}$ −29.6 (C 1.066 DCM). ^1^H-NMR (400 MHz, CDCl_3_) δ 7.47 (s, 1H, NCH), 6.67 (ddd, *J* = 5.7, 2.6, 1.3 Hz, 1H, HC=CCO), 5.11 (s, 1H^a^, =CH_2_), 5.05 (s, 1H^b^, =CH_2_), 5.02 (d, *J* = 15.4 Hz, 1H, H_2_C=C-CH_2_N), 4.92 (d, *J* = 15.2 Hz, 1H, H_2_C=C-CH_2_N), 3.77 (s, 2H, CH_2_N(CH_2_)_4_), 2.64–2.52 (m, 2H, CH_2_C=O, 4H, N(CH_2_CH_2_)_2_), 2.47 (m, 1H, CH in carbocycle), 2.36 (m, 1H, CH_2_ in carbocycle), 2.29–2.17 (m, 1H, CH_2_ in carbocycle), 1.83–1.76 (m, 4H, N(CH_2_CH_2_)_2_), 1.76–1.71 (m, 3H, CH_3_). ^13^C-NMR (101 MHz, DMSO) δ 198.6 (C=O), 146.7, 145.2, 143.8, 135.8, 122.2, 115.5, 54.2, 54.0, 51.0, 42.8, 38.3, 31.3, 23.6, 15.7. HRMS (ESI) *m*/*z*: [M + H]^+^ Calcd for C_17_H_25_N_4_O^+^ 301.2028, Found 301.2026.

*(5R)-2-methyl-5-(3-{4-[(piperidin-1-yl)methyl]-1H-1,2,3-triazol-1-yl}prop-1-en-2-yl)cyclohex-2-en-1-one* (**4d**). Yield 79%, m.p. 80–81 °C, ${\left[\alpha \right]}_{D}^{20}$ −35.04 (C 1.036 DCM). ^1^H-NMR (400 MHz, CDCl_3_) δ 7.46 (s, 1H, NCH), 6.67 (ddd, *J* = 5.8, 2.6, 1.4 Hz, 1H, HC=CCO), 5.12 (s, 1H^a^, H_2_C=), 5.06 (s, 1H^b^, H_2_C=), 5.02 (d, *J* = 15.4 Hz, 1H^a^, H_2_C=C-CH_2_N), 4.92 (d, *J* = 15.2 Hz, 1H^b^, H_2_C=C-CH_2_N), 3.64 (s, 2H, CH_2_N(CH_2_)_5_), 2.65–2.30 (m, 8H, CH_2_C=O, N(CH_2_CH_2_)_2_CH_2_, CH and CH_2_ in carbocycle), 2.29–2.16 (m, 1H CH_2_ in carbocycle), 1.75 (dt, *J* = 2.3, 1.2 Hz, 3H, CH_3_), 1.61–1.52 (m, 4H, N(CH_2_CH_2_)_2_CH_2_), 1.46–1.36 (m, 2H, N(CH_2_CH_2_)_2_CH_2_). ^13^C-NMR (75 MHz, DMSO/CCl_4_—1/3) δ 196.4 (C=O), 145.6, 143.8, 142.9, 134.4, 122.7, 113.3, 53.4, 52.5, 42.1, 37.9, 30.4, 25.3, 23.7, 15.1. HRMS (ESI) *m*/*z*: [M + H]^+^ Calcd for C_18_H_27_N_4_O^+^ 315.2185, Found 315.2182.

*(5R)-2-methyl-5-(3-{4-[(azepan-1-yl)methyl]-1H-1,2,3-triazol-1-yl}prop-1-en-2-yl)cyclohex-2-en-1-one* (**4e**). Yield 81%, m.p. 89–90 °C, ${\left[\alpha \right]}_{D}^{20}\;$ −21.6 (C 1.000 DCM). ^1^H-NMR (400 MHz, CDCl_3_) δ 7.46 (s, 1H, NCH), 6.66 (ddd, *J* = 5.7, 2.6, 1.4 Hz, 1H, HC=CCO), 5.11 (s, 1H^a^, H_2_C=), 5.05 (s, 1H^b^, H_2_C=), 5.02 (d, *J* = 15.6 Hz, 1H^a^, CH_2_=C-CH_2_N), 4.92 (d, *J* = 15.2 Hz, 1H^b^, CH_2_=C-CH_2_N), 3.79 (s, 2H, CH_2_N(CH_2_)_6_), 2.68–2.60 (m, 4H, N(CH_2_CH_2_CH_2_)_2_), 2.60–2.29 (m, 4H, CH and CH_2_ in carbocycle), 2.29–2.04 (m, 1H, CH_2_ in carbocycle), 1.74 (dt, *J* = 2.4, 1.2 Hz, 3H, CH_3_), 1.68–1.51 (m, 8H, N(CH_2_CH_2_CH_2_)_2_). ^13^C-NMR (101 MHz, CDCl_3_) δ 198.6 (C=O), 146.9, 145.3, 143.8, 135.8, 122.4, 115.4, 55.6, 54.0, 53.6, 42.8, 38.3, 31.3, 28.1, 27.0, 15.7. HRMS (ESI) *m*/*z*: [M + H]^+^ Calcd for C_19_H_29_N_4_O^+^ 329.2341, Found 329.2339.

*1-[(1-{2-[(1R)-4-methyl-5-oxocyclohex-3-en-1-yl]prop-2-en-1-yl}-1H-1,2,3-triazol-4-yl)methyl]pyrrolidine-2,5-dione* (**4f**). Yield 65%, m.p. 127–128 °C, ${\left[\alpha \right]}_{D}^{20}\;$ −23.2 (C 1.61 DCM). ^1^H-NMR (400 MHz, CDCl_3_) δ 7.56 (s, 1H, NCH), 6.68 (ddd, *J* = 5.6, 2.6, 1.4 Hz, 1H, HC=CCO), 5.13 (s, 1H^a^, H_2_C=), 5.06 (s, 1H^b^, H_2_C=), 5.00 (d, *J* = 15.2 Hz, 1H^a^, CH_2_=C-CH_2_N), 4.90 (d, *J* = 15.2 Hz, 1H^b^, CH_2_=C-CH_2_N), 4.79 (s, 2H, CH_2_N-C=O), 2.74 (s, 4H, CH_2_C(O)N), 2.61–2.43 (m, 3H, CH and CH_2_ in carbocycle), 2.37–2.18 (m, 2H, CH and CH_2_ in carbocycle), 1.76–1.74 (m, 3H, CH_3_). ^13^C-NMR (101 MHz, CDCl_3_) δ 198.5(C=O), 176.7(C=O), 144.9, 143.8, 142.6, 135.8, 123.2, 115.8, 54.1, 42.8, 38.2, 33.8, 31.2, 29.8, 28.3, 15.7. HRMS (ESI) *m*/*z*: [M + H]^+^ Calcd for C_17_H_21_N_4_O_3_^+^ 329.1614, Found 329.1610.

*1-[(1-{2-[(1R)-4-methyl-5-oxocyclohex-3-en-1-yl]prop-2-en-1-yl}-1H-1,2,3-triazol-4-yl)methyl]-1H-indole-2,3-dione* (**4g**). Yield 63%, m.p. 144–145 °C, ${\left[\alpha \right]}_{D}^{20}\;$ −23.1 (C 1.6(6) DCM). ^1^H-NMR (400 MHz, CDCl_3_) δ 7.58 (s, 1H, NCH), 7.53–7.44 (m, 2H_arom_), 7.16 (d, *J* = 8.3 Hz, 1H_arom_), 7.02 (t, *J* = 7.5 Hz, 1H_arom_), 6.57 (ddd, *J* = 5.7, 2.6, 1.3 Hz, 1H, HC=CCO), 5.03 (s, 1H^a^, H_2_C=), 4.95 (s, 1H^b^, H_2_C=), 4.92 (s, 2H, CH_2_N-C=O), 4.91 (d, *J* = 15.6 Hz, 1H^a^, CH_2_=C-CH_2_N), 4.85 (d, *J* = 15.2 Hz, 1H^b^, CH_2_=C-CH_2_N), 2.54–2.47 (m, 1H^a^, CH_2_ in carbocycle), 2.47–2.40 (m, 1H^b^, CH_2_ in carbocycle), 2.40–2.30 (m, 1H, CH in carbocycle), 2.29–2.19 (m, 1H^a^, CH_2_ in carbocycle), 2.19–2.07 (m, 1H^b^, CH_2_ in carbocycle), 1.69–1.58 (m, 3H, CH_3_). ^13^C-NMR (101 MHz, CDCl_3_) δ 198.4(C=O), 183.1(C=O), 158.0, 150.2, 144.6, 143.7, 142.3, 138.7, 135.7, 125.4, 124.2, 123.1, 117.6, 115.9, 111.5, 54.1, 42.7, 38.2, 35.5, 31.1, 15.7. HRMS (ESI) *m*/*z*: [M + H]^+^ Calcd for C_18_H_24_N_5_O_3_^+^ 377.1614, Found 377.1609.

*2-[(1-{2-[(1R)-4-methyl-5-oxocyclohex-3-en-1-yl]prop-2-en-1-yl}-1H-1,2,3-triazol-4-yl)methyl]-1H-isoindole-1,3(2H)-dione* (**4h**). Yield 84%, m.p. 132 °C, ${\left[\alpha \right]}_{D}^{20}\;$ –21.3 (C 1.6(6) DCM). ^1^H-NMR (400 MHz, CDCl_3_) δ 7.93–7.80 (m, 2H_arom_), 7.80–7.66 (m, 2H_arom_), 7.59 (s, 1H, NCH), 6.73–6.59 (m, 1H, HC=CCO), 5.11 (s, 1H^a^, H_2_C=), 5.05 (s, 1H^b^, H_2_C=), 5.00 (d, *J* = 14.6 Hz, 1H^a^, CH_2_=C-CH_2_N), 4.99 (s, 1H, CH_2_N-C=O), 4.90 (d, *J* = 15.2 Hz, 1H^b^, CH_2_=C-CH_2_N), 2.66-2.51 (m, 2H, CH_2_ in carbocycle), 2.47 (m, 1H, CH in carbocycle), 2.36 (m, 1H, CH_2_ in carbocycle), 2.28–2.12 (m, 1H, CH_2_ in carbocycle), 1.82–1.63 (m, 3H). ^13^C-NMR (101 MHz, CDCl_3_) δ 198.6(C=O), 167.8(C=O), 144.9, 143.8, 135.8, 134.5, 134.3, 132.1, 123.6, 123.0, 115.7, 54.1, 42.7, 38.2, 33.2, 31.2, 15.7. HRMS (ESI) *m*/*z*: [M + H]^+^ Calcd for C_21_H_21_N_4_O_3_^+^ 377.1614, Found 377.1609.

*(5R)-5-(3-{4-[(1H-benzotriazol-1-yl)methyl]-1H-1,2,3-triazol-1-yl}prop-1-en-2-yl)-2-methylcyclohex-2-en-1-one* (**4i**). Yield 76%, m.p. 104–105 °C, ${\left[\alpha \right]}_{D}^{20}\;$ −24.0 (C 1.67 DCM). ^1^H-NMR (400 MHz, CDCl_3_) δ 8.04 (dt, *J* = 8.4, 0.9 Hz, 1H_arom_), 7.68 (dt, *J* = 8.4, 0.9 Hz, 1H_arom_), 7.50 (s, *J* = 3.2 Hz, 1H, NCH), 7.47 (ddd, *J* = 8.3, 7.0, 1.0 Hz, 1H_arom_), 7.39–7.34 (m, 1H_arom_), 6.62 (ddd, *J* = 5.6, 2.7, 1.3 Hz, 1H, HC=CCO), 5.97 (s, 2H, CH_2_N), 5.09 (s, 1H^a^, H_2_C=), 5.00 (s, 1H^b^, H_2_C=), 4.96 (d, *J* = 15.4 Hz, 1H^a^, CH_2_=C-CH_2_N), 4.88 (d, *J* = 15.2 Hz, 1H^b^, CH_2_=C-CH_2_N), 2.59–2.47 (m, 2H, CH_2_ in carbocycle), 2.45–2.30 (m, 2H, CH_2_ in carbocycle), 2.30–2.11 (m, 1H, CH_2_ in carbocycle), 1.75–1.72 (m, 3H, CH_3_). ^13^C-NMR (101 MHz, CDCl_3_) δ 198.3(C=O), 144.6, 143.6, 142.8, 135.8, 127.9, 124.3, 123.0, 120.0, 116.0, 110.1, 54.3, 42.7, 38.2, 31.1, 29.8, 15.7. HRMS (ESI) *m*/*z*: [M + H]^+^ Calcd for C_19_H_21_N_6_O^+^ 349.1777, Found 349.1773.

The ^1^H-NMR and ^13^C-NMR for all the synthesized compounds as well as details of fluorescence experiments and BSA binding are available in Appendix A.

## 4. Conclusions

Efficient synthesis of 1,2,3-triazolyl conjugates of various amines with carvone was developed. Chlorination of carvone followed by the reaction with sodium azide open excess to 10-azidocarvone. Its CuAAC with propargylated amines can be performed as a one-pot procedure to give target products in up to 84% isolated yield. The prepared conjugates demonstrated high antioxidant activity and spontaneously undergo hydrophobic reversible interaction with serum albumin.

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
