# Peer review of "Synthesis of Carvone-Derived 1,2,3-Triazoles Study of Their Antioxidant Properties and Interaction with Bovine Serum Albumin"

_molecules, 2018, doi:10.3390/molecules23112991_

Round 1

Reviewer 1 Report

The manuscript by V. Nenajdenko et al. describes the synthesis of a series of carvone-derived materials functionalized with tertiary amino/amido groups linked via 1,2,3-triazole unit. The title compounds were prepared in a two-step one-pot protocol starting with known 10-chlorocarvone by nucleophilic displacement of chlorine using sodium azide followed by Huisgen-Sharpless-Meldal click cycloaddition of the initially formed azide and selected terminal acetylenes. The structures of the final products, and of some intermediate compounds, were fully confirmed by spectroscopic methods supplemented with high resolution MS. The target molecules were tested as potential antioxidants by using PNDMA as a competitive acceptor (bleaching reference) of the UV-generated hydroxyl radical to show mentionable activity comparable to that of ascorbic acid used as a control. This referee recommend publication after the following points were addressed:

The influence of both triazole and amine/amide groups on the antioxidant properties of the target compounds is not clear as the Michael acceptor functionality (and/or methylidene groups) present in the carvone-derived unit seems crucial for radical trapping. An additional experiment testing starting terpene (carvone) would shed some light on this issue, and the comparable analysis should be provided by the Authors.

Graphical abstract: the presented structure of albumin is readily available on the Internet (e.g. https://www.sasbdb.org/data/SASDBT4/) and may by copyrighted. An agreement by the other parties to reproduce this picture may by required.

It is not clear why the lower loading of the Cu-catalyst increases the chemical yield (table 1, entries 5 and 6). Please briefly explain in the text.

Further minor corrections/comments:

Abstract, second line: add ‘h’ to read ‘chlorination’

Page 2, first line. The statement about annual sales should be supplemented with appropriate literature reference.

Page 2, second paragraph, first sentence:      add ‘the’ to read ‘the simplest’

Page 3, top: the sentence ‘Ribavirin [24,25] – a drug …’ needs predicate.

Page 3, results & discussion, first paragraph: use bolded fonts for numbering of compounds 4a-i and for derivatives 3.

Figures 4-8, and Fig 9; please specify time (min, sec ?) and concentration (mol/L) units in figures, respectively, and use bolded fonts for numbering of compounds (caption).

Conclusion: change to read ‘1,2,3-triazolyl’

Experimental part, compound 3g, 13C-NMR: based on the chemical shifts and signals intensities the absorptions at 137.5 and 117.2 ppm should be attributed to CH(arom) and i-C(neighbouring the CO group) atoms, respectively, while it is the opposite in the text.

Author Response

Thank you very much for valuble rmarks concerning our manuscript.

We made the corresponding corrections in our revision.

ALL your recommendation were taken into account.

The corresponding laces are marked in yellow.

Please find below point-by-point responce.

The influence of both triazole and amine/amide groups on the antioxidant properties of the target compounds is not clear as the Michael acceptor functionality (and/or methylidene groups) present in the carvone-derived unit seems crucial for radical trapping. An additional experiment testing starting terpene (carvone) would shed some light on this issue, and the comparable analysis should be provided by the Authors.

The corresponding sentence was added.

Graphical abstract: the presented structure of albumin is readily available on the Internet (e.g. https://www.sasbdb.org/data/SASDBT4/) and may by copyrighted. An agreement by the other parties to reproduce this picture may by required.

This picture is very common and no citation is given as a rule. 

It is not clear why the lower loading of the Cu-catalyst increases the chemical yield (table 1, entries 5 and 6). Please briefly explain in the text.

Corrected

Further minor corrections/comments:

Abstract, second line: add ‘h’ to read ‘chlorination’

Corrected

Page 2, first line. The statement about annual sales should be supplemented with appropriate literature reference.

Corrected

Page 2, second paragraph, first sentence:      add ‘the’ to read ‘the simplest’

Corrected

Page 3, top: the sentence ‘Ribavirin [24,25] – a drug …’ needs predicate.

Corrected

Page 3, results & discussion, first paragraph: use bolded fonts for numbering of compounds 4a-i and for derivatives 3.

Corrected

Figures 4-8, and Fig 9; please specify time (min, sec ?) and concentration (mol/L) units in figures, respectively, and use bolded fonts for numbering of compounds (caption).

Corrected

Conclusion: change to read ‘1,2,3-triazolyl’

Corrected

Experimental part, compound 3g, 13C-NMR: based on the chemical shifts and signals intensities the absorptions at 137.5 and 117.2 ppm should be attributed to CH(arom) and i-C(neighbouring the CO group) atoms, respectively, while it is the opposite in the text.

Corrected

Reviewer 2 Report

Dear authors,

You present here the synthesis of 9 carvone-1,2,3-triazoles, their antioxidant activity and their interaction with albumin. The paper requires major English corrections, there are phrases without logic (lines 48-49), without verbs (lines 73-74). Please check the instructions for authors of this journal: the references are written after the mark of the end of the phrase. 

The figures 1-3 and tables 1 and 2 are not cited in the text. Rephrase line 109: "Due to the best water-solubility,...".and also line 143 "open excess...", maybe you want to say "access"?

You say that the compounds were obtained with yield up to 91%. I checked the experimental part and the highest yield for compounds 4 a-i was 84%. 

Author Response

Thank you very much for valuble rmarks concerning our manuscript.

We made the corresponding corrections in our revision.

ALL your recommendation were taken into account.

The corresponding laces are marked in yellow.

Please find below point-by-point responce.

You present here the synthesis of 9 carvone-1,2,3-triazoles, their antioxidant activity and their interaction with albumin. The paper requires major English corrections, there are phrases without logic (lines 48-49), without verbs (lines 73-74). Please check the instructions for authors of this journal: the references are written after the mark of the end of the phrase. 

corrected

The figures 1-3 and tables 1 and 2 are not cited in the text. 

corrected

Rephrase line 109: "Due to the best water-solubility,...".and also line 143 "open excess...", maybe you want to say "access"?

corrected

You say that the compounds were obtained with yield up to 91%. I checked the experimental part and the highest yield for compounds 4 a-i was 84%. 

corrected

Reviewer 3 Report

The manuscript entitled “Synthesis of Carvone derived 1,2,3-Triazoles. Study of their Antioxidant Properties and Interaction with Bovine Serum Albumin” give a poor scientific contribution from my point of view.  

The reactions described in the manuscript has been extensively studied and similar conditions has been employed (e.g. Synlett, 2009, 1453-1456; Guangdong Huagong, 39(8), 6-7; 2012), therefore, the contribution to synthetic organic chemistry is mainly the use of carvone as a substrate.

On the other hand, from a medicinal chemistry point of view, the contribution is also poor, the molecules lack a rational design and the biological assays are poorly supported by the introduction. The report of antioxidant activity and albumin interaction should be complemented with other assays and the design of the compounds should be properly justified.

In general, I found this manuscript of low interest and should be improved with additional experiments.

Author Response

Thank you very much for valuble remarks concerning our manuscript.

We made the corresponding corrections in our revision.

ALL your recommendation were taken into account.

The corresponding places are marked in yellow.

The main goal of this study was to elaborate synthesis of such carvone conjugates as a proof of principle. In future we are going to have more detailed study directed to their medicinal chemistry.

Round 2

Reviewer 2 Report

Dear authors,

I observed that you made the changes suggested.

Author Response

Thank you very much!

All your comments were used for revision.

Reviewer 3 Report

The authors made some corrections on the manuscript following the recommendations, however, some minor errors are still pending.

Line 82, page 3. Scaffolds should be moieties.

Scheme 1. Up to 91% should be 84.

Line 148, page 7. In the conclusions section Clorination should be chlorination.

Line 150, page 7. In the conclusions section, again up to 91% should be 84.

Author Response

Thank you very much for your valuable remarks!

Line 82, page 3. Scaffolds should be moieties.

corrected

Scheme 1. Up to 91% should be 84.

corrected

Line 148, page 7. In the conclusions section Clorination should be chlorination.

corrected

Line 150, page 7. In the conclusions section, again up to 91% should be 84.

corrected